# Regret Bounds for Learning State Representations in Reinforcement Learning

**Ronald Ortner**
Montanuniversität Leoben
rortner@unileoben.ac.at

**Matteo Pirotta**
Facebook AI Research
pirotta@fb.com

**Ronan Fruit**
Sequel Team – INRIA Lille
ronan.fruit@inria.fr

**Alessandro Lazaric**
Facebook AI Research
lazaric@fb.com

**Odalric-Ambrym Maillard**
Sequel Team – INRIA Lille
odalric.maillard@inria.fr

## Abstract

We consider the problem of online reinforcement learning when several state representations (mapping histories to a discrete state space) are available to the learning agent. At least one of these representations is assumed to induce a Markov decision process (MDP), and the performance of the agent is measured in terms of cumulative regret against the optimal policy giving the highest average reward in this MDP representation. We propose an algorithm (UCB-MS) with $\widetilde{O}(\sqrt{T})$ regret in any communicating MDP. The regret bound shows that UCB-MS automatically adapts to the Markov model and improves over the currently known best bound of order $\widetilde{O}(T^{2/3})$.

## 1 Introduction

In reinforcement learning, an agent aims to learn a task while interacting with an unknown environment. We consider online learning (i.e., non-episodic) problems where the agent has to trade off the *exploration* needed to collect information about rewards and dynamics and the *exploitation* of the information gathered so far. In this setting, it is commonly assumed that the agent knows a suitable *state representation* which makes the process on the state space Markovian. However, designing such a representation is often highly non-trivial since many "reasonable" representations may lead to non-Markovian models.

The task of selecting or designing a (suitable and compact) state representation of a dynamical system is a well-known problem in engineering, especially in robotics. This setting has received a lot of attention in recent years due to the growing number of applications using images or videos as observations [e.g., 1, 2, 3]. It is possible to come up with different approaches for extracting features from such high-dimensional observation spaces, but not all of them describe the problem well or exhibit Markovian dynamics. Indeed, the Markovian assumption that transitions and rewards are independent of history is frequently violated in real-world applications where there is often rather a dependence on the last $k > 1$ observations. To deal with this scenario Markov models have been extended from first-order models to $k$th-order models. The problem of selecting the right order of the model is a special case of the selection of the correct state representation. In both cases, the goal is to perform as well as when the true order or compact features of the Markov model are known. For more details and further examples we refer to [4, 5, 6].

We consider the following setting that was introduced by Hutter [7], where it was called *feature reinforcement learning*. The agent is provided with a finite set $\Phi$ of representations mapping histories (sequences of actions, observations, and rewards) to a *finite* set of states, such that at least one model $\phi^\circ \in \Phi$ induces a Markov decision process (MDP) [8]. The goal of the agent is to learn to solve the

task under an appropriate representation. The ability of testing and quickly discarding non-Markovian representations (not compatible with the observed dynamics) is fundamental for learning efficiently. This efficiency is measured in terms of cumulative *regret*, which compares the reward collected by the learner to the one of an agent knowing the Markovian representation and playing the associated optimal policy (i.e., the policy giving the highest average reward).

This problem was initially studied by Maillard et al. [4]. Given a finite set of representations $\Phi$, after $T$ steps the regret of the Best Lower Bound (BLB) algorithm w.r.t. any optimal policy associated to a Markov model is upper bounded by $\widetilde{O}(\sqrt{|\Phi|}T^{2/3})$. The BLB algorithm is based on random exploration of the models and uses properties of UCRL2 [9] —an efficient algorithm for exploration-exploitation in communicating MDPs— to control the amount of time spent in non-Markovian models. BLB requires to estimate the diameter [9] of the true MDP, which leads to an additional additive term in the regret bound that may be exponential in the true diameter. BLB was successively extended by Nguyen et al. [6] to the case of countably infinite sets of models. The suggested IBLB algorithm removes the guessing of the diameter —thus avoiding the additional exponential term in the regret— but its regret bound is still of order $T^{2/3}$. For the Optimistic Model Selection (OMS) algorithm [5] a regret bound of $\widetilde{O}(\sqrt{|\Phi|T})$ has been claimed that matches the optimal dependence in terms of $T$. However, algorithm and analysis were based on the REGAL.D algorithm [10], and recently it has been pointed out that the proof of the regret bound for REGAL.D contains a mistake that invalidates also the result for OMS, see App. A of [11]. Accordingly, it still has been an open question whether it is possible to achieve regret bounds of order $\sqrt{T}$ in the considered setting.

In this paper we introduce UCB-MS, an optimistic elimination algorithm that performs efficient exploration of the representations. For this algorithm we prove a regret bound of order $\widetilde{O}(\sqrt{|\Phi|T})$. Our algorithm as well as our results are based on and generalize the regret bounds achieved for the UCRL2 algorithm in [9]. In particular, if $\Phi$ consists of a single Markov model we obtain the same regret bound as for UCRL2. UCB-MS employs optimism to choose a model from $\Phi$. To avoid suffering too large regret from choosing a non-Markov model, the collected rewards are compared to regret bounds that are known to hold for Markov models. If a model fails to give sufficiently high reward, it is eliminated. On the other hand, UCB-MS is happy to employ a non-Markov model as long as it gives as much reward as it would be expected from a corresponding Markov model.

While UCB-MS shares some basic ideas with BLB and OMS, it is simpler than OMS, however recovers the same regret bounds that have been claimed for OMS. As BLB, UCB-MS has to guess the diameter, however the guessing scheme we employ comes at little cost w.r.t. regret and in particular does not cause any additive constants in the bounds that are exponential in the diameter. We also show how to modify the guessing scheme to guess diameter and the size of the state space of the Markov model $\phi^\circ$ at the same time. Last but not least, we introduce the notion of the *effective* size $S_\Phi$ of the state space induced by the model set $\Phi$ and give regret bounds in terms of $S_\Phi$. This yields improvements depending on the structure of $\Phi$ (like for hierarchical models).

**Overview.** We start with a detailed description of the learning setting in the following section. In Section 3, we give some preliminaries concerning the UCRL2 algorithm. Our UCB-MS algorithm is introduced in Section 4 where we also give the regret analysis in case the diameter of the underlying Markov model is known. The following Section 5 shows how to guess the diameter otherwise. Section 6 gives some further improvements and also introduces the notion of *effective* state space.

## 2 Setting

The details of the considered learning setting are as follows. At each time step $t = 1, 2, \ldots$, the learner receives an initial observation $o_t$ and has to choose an action $a_t$ from a finite set of actions $\mathcal{A}$. In return, the learner receives a reward $r_t$ taken from $\mathcal{R} = [0, 1]$ and the next observation $o_{t+1}$.

We denote by $\mathcal{O}$ the set of observations and define the history $h_t$ up to step $t$ as the sequence $o_1, a_1, r_1, o_2, \ldots, a_t, r_t, o_{t+1}$ of observations, actions and rewards. The set $\mathcal{H}_t := \mathcal{O} \times (\mathcal{A} \times \mathcal{R} \times \mathcal{O})^{t-1}$ contains all possible histories up to step $t$ and we set $\mathcal{H} := \bigcup_{t \geq 1} \mathcal{H}_t$ to be the set of all possible histories.

### 2.1 Models and MDPs

A *state-representation model* (in the following short: *model*) $\phi$ is a function that maps histories to states, that is, $\phi : \mathcal{H} \to \mathcal{S}_\phi$. If a model $\phi$ induces a *Markov decision process (MDP)* we call it a

*Markov model.* An MDP has the Markov property that any time $t$, the probability of reward $r_t$ and next state $s_{t+1} = \phi(h_{t+1})$, given the past history $h_t$, only depends on the current state $s_t = \phi(h_t)$ and the chosen action $a_t$, i.e., $P(s_{t+1}, r_t | h_t, a_t) = P(s_{t+1}, r_t | s_t, a_t)$. We assume that for MDPs this probability is also independent of $t$.

Usually, an MDP $M$ with state space $\mathcal{S}$ and action space $\mathcal{A}$ is denoted as a tuple $M = (\mathcal{S}, \mathcal{A}, r, p)$, where $r(s, a)$ is the mean reward and $p(s' | s, a)$ the probability of a transition to state $s' \in \mathcal{S}$ when choosing action $a \in \mathcal{A}$ in state $s \in \mathcal{S}$. MDPs are called *communicating* if for any two states $s, s'$ it is possible to reach $s'$ from $s$ with positive probability by choosing appropriate actions. The smallest expected time it takes to connect any two states is called the *diameter $D$* of the MDP, cf. [9]. In communicating MDPs, the optimal average reward $\rho^*$ is independent of the initial state and will be achieved by a stationary deterministic policy $\pi^* \in \Pi^{SD}$ that maps states to actions.

For a Markov model $\phi$, we write $M(\phi)$ for the induced MDP. Its diameter and optimal average reward will be denoted as $D(\phi)$ and $\rho^*(\phi)$, respectively. For all models $\phi$, we abbreviate $S_\phi := |\mathcal{S}_\phi|$.

## 2.2 Problem setting

The learning setting we consider is the following. As already described before, the learner chooses actions $a_t$ and obtains a reward $r_t$ and an observation $o_{t+1}$ in return. We assume that the learner has a finite set $\Phi$ of models at her disposal and at least one model $\phi^\circ$ in $\Phi$ is a Markov model. The goal is to provide algorithms that perform well with respect to the optimal policy $\pi^*$ in the MDP $M(\phi^\circ)$, that is, the optimal strategy when the Markov model and the induced underlying MDP are completely known. Accordingly, the performance of a learning algorithm will be measured by considering its *regret* after any $T$ steps defined as (cf. [9, 10, 4])

$$T\rho^*(\phi^\circ) - \sum_{t=1}^{T} r_t \,,$$

where $r_t$ is the reward received by the learning algorithm at step $t$.

# 3 UCRL2 Preliminaries

The algorithm we propose is based on the UCRL2 algorithm of [9]. Accordingly, we start with some respective preliminaries.

UCRL2 is an algorithm that generalizes the *optimism in the face of uncertainty* idea of UCB [12] from the bandit setting to reinforcement learning in MDPs. In the following, we assume an underlying MDP $M$ with $S$ states, $A$ actions, and diameter $D$. The UCRL2 algorithm uses confidence intervals to define a set of plausible MDPs $\mathcal{M}$. That is, acting in the unknown MDP $M$, UCRL2 maintains estimates $\hat{r}(s, a)$ and $\hat{p}(\cdot | s, a)$ of rewards and transition probabilities, respectively. The set $\mathcal{M}_t$ of plausible MDPs at step $t$ contains all MDPs with rewards $\widetilde{r}(s, a)$ and transition probabilities $\widetilde{p}(\cdot | s, a)$ satisfying[1]

$$
\left| \hat{r}(s, a) - \widetilde{r}(s, a) \right| \leq \sqrt{\frac{7 \log(4SAt^3/\delta)}{2N(s,a)}}, \tag{1}
$$

$$
\left\| \hat{p}(\cdot | s, a) - \widetilde{p}(\cdot | s, a) \right\|_1 \leq \sqrt{\frac{14S \log(4At^3/\delta)}{2N(s,a)}}, \tag{2}
$$

where $N(s, a)$ denotes the number of times $a$ has been chosen in $s$ (and is set to 1, if $a$ has not been chosen in $s$ so far). The true MDP $M$ is in $\mathcal{M}$ with high probability.

**Lemma 1** (Lemma 17 in the appendix of [9][2]). *With probability at least $1 - \frac{\delta}{30t^8}$, at step $t$ the true MDP $M$ is contained in the set $\mathcal{M}_t$.*

The UCRL2 algorithm proceeds in episodes $k = 1, 2, \ldots$. In each episode $k$ starting at step $t_k$ the algorithm plays a fixed policy $\widetilde{\pi}_k$, which is chosen to maximize the optimal average reward of an MDP in $\mathcal{M}_k := \mathcal{M}_{t_k}$. That is, writing $\rho(\pi, M)$ for the average reward of policy $\pi$ in MDP $M$ we

set $\widetilde{\rho}_k := \max_{\pi, M \in \mathcal{M}_k}\{\rho(\pi, M)\} = \rho(\widetilde{\pi}_k, \widetilde{M}_k)$, where $\widetilde{M}_k$ is an optimistic MDP chosen from $\mathcal{M}_k$ to maximize $\widetilde{\rho}_k$. As the true MDP $M$ is in $\mathcal{M}_k$ with high probability, we also have that $\widetilde{\rho}_k \geq \rho^*$.

Let $v_k(s, a)$ denote the number of times $a$ has been chosen in $s$ in episode $k$, while $N_k(s, a)$ denotes the number of times $a$ has been chosen in $s$ before episode $k$ (i.e., in episodes 1 to $k - 1$). If there were no visits in $(s, a)$ before episode $k$, then $N_k(s, a)$ is set to 1. Episode $k$ is terminated by UCRL2 when a state $s$ is reached in which $v_k(s, \widetilde{\pi}_k(s)) = N_k(s, \widetilde{\pi}_k(s))$.

One can show the following upper bound on the regret of UCRL2.

**Theorem 2** (Theorem 2 of [9]). *With probability $1 - \delta$, the regret of UCRL2 after any $T$ steps is bounded by*

$$34DS\sqrt{AT \log\left(\tfrac{2T^3}{\delta}\right)}.$$

The bound is based on an episode-wise decomposition of the regret, which we will use for our algorithm. Let $T_k$ be the (current) length of episode $k$. In the following, we abuse notation for $T_k$ as well as for $v_k(s, a)$ by using the same notation for the number of steps in a terminated episode as well as for the current number of steps in an ongoing episode. Further, recall that $t_k$ denotes the time step when episode $k$ starts. The regret of UCRL2 in any episode $k$ is bounded as follows.

**Lemma 3.** *Consider an arbitrary episode $k$ started at step $t_k$. With probability $1 - \frac{\delta}{2t_k^2}$, the regret of UCRL2 at each step $T_k$ in this episode is bounded by*

$$\left(2D\sqrt{14S \log\left(\tfrac{4At_k^3}{\delta}\right)} + 2\right) \sum_{s,a} \frac{v_k(s,a)}{\sqrt{N_k(s,a)}} + 4D\sqrt{T_k \log\left(\tfrac{16t_k^2 T_k}{\delta}\right)} + D.$$

*Proof.* The bound in Lemma 3 is not explicitly stated for single episodes in [9] but easily follows from equations (8), (9), (15)–(17), and the argument given before equation (18), choosing confidence $\delta/t^2$ instead of $\delta$. For the sake of completeness, we give a brief proof sketch.

First, by replacing the random rewards by the mean rewards $r(s, a)$, the regret $\Delta_k$ of episode $k$ can be bounded by (cf. Eq. 8 in [9])

$$\Delta_k \leq \sum_{s,a} v_k(s,a)(\rho^* - r(s,a)) + \sqrt{\tfrac{5}{8}T_k \log\left(\tfrac{16t_k^2 T_k}{\delta}\right)}. \tag{3}$$

Let $\tilde{r}$ and $\tilde{p}$ denote the rewards and transition probabilities in the optimistic MDP $\widetilde{M}_k$. Then the difference in the sum of (3) can be bounded and split up as

$$\rho^* - r(s,a) \leq \widetilde{\rho}_k - r(s,a) \leq (\widetilde{\rho}_k - \tilde{r}(s,a)) + (\tilde{r}(s,a) - r(s,a)), \tag{4}$$

where the last term is controlled by the confidence intervals in (1), cf. Eq. (15) in [9]. The other term can be written as $\widetilde{\rho}_k - \tilde{r}(s,a) = \sum_{s'} \tilde{p}(s'|s,a)w(s') - w(s)$ for the shifted value vector $w$ (cf. p. 1576 of [9]) so that splitting up again one has (cf. Eq. 16 in [9])

$$\widetilde{\rho}_k - \tilde{r}(s,a) = \sum_{s'}\big(\tilde{p}(s'|s,a) - p(s'|s,a)\big)w(s') + \left(\sum_{s'}\big(\tilde{p}(s'|s,a) - w(s)\big)\right). \tag{5}$$

The first term is handled by the confidence intervals in (2) and the fact the $w(s) \leq D$ (cf. Eq. 17 in [9]), while the second term can be written as martingale difference sequence and bounded by $D\sqrt{\tfrac{5}{2}T_k \log\left(\tfrac{16t_k^2 T_k}{\delta}\right)} + D$ using Azuma-Hoeffding (cf. Eq. 18 in [9]). Finally taking into account an additional regret term of $\sqrt{T_k}$ caused by failing confidence intervals (cf. Eq. 9 in [9]) and combining (3)–(5) gives the claimed bound, where the first term stems from the confidence intervals (1) and (2). $\qquad\square$

## 4 The UCB-MS Algorithm

Now let us turn to the state representation learning setting introduced in Section 2. We start with the simpler case when an upper bound $\bar{D}$ on the diameter $D := D(\phi^\circ)$ of the Markov model $\phi^\circ$ is known (i.e., $\bar{D} \geq D$). The case when no bound on the diameter is known is dealt with in Section 5.

---

**Algorithm 1** UCB-Model Selection (UCB-MS)

---

**Input:** set of models $\Phi$, confidence parameter $\delta \in (0, 1)$, upper bound $\bar{D}$ on diameter
**Initialization:** Let $t := 1$ be the current time step.
**for** episodes $k = 1, 2, \ldots$ **do**

    Let $t_k := t$ be the initial step of the current episode $k$, and let $\mathcal{M}_{t,\phi}$ be the set of *plausible* MDPs defined via the confidence intervals (1) and (2) for the estimates so far.

    $\triangleright$ For each $\phi \in \Phi$, use Extended Value Iteration (EVI) to compute an optimistic MDP $\widetilde{M}_{k,\phi}$ in $\mathcal{M}_{t,\phi}$ and a (near-)optimal policy $\widetilde{\pi}_{k,\phi}$ on $\widetilde{M}_{k,\phi}$ with approximate average reward $\widetilde{\rho}_{k,\phi}$.

    $\triangleright$ Choose model $\phi_k \in \Phi$ such that

$$\phi_k = \operatorname*{argmax}_{\phi \in \Phi} \left\{ \widetilde{\rho}_{k,\phi} \right\}, \tag{8}$$

    and set $\widetilde{\rho}_k := \widetilde{\rho}_{k,\phi_k}$, $\widetilde{\pi}_k := \widetilde{\pi}_{k,\phi_k}$, and $\mathcal{S}_k := \mathcal{S}_{\phi_k}$.

    $\triangleright$ Repeat till termination of the current episode $k$:
        $\circ$ Choose action $a_t := \pi_k(s_t)$, get reward $r_t$ and observe next state $s_{t+1} \in \mathcal{S}_k$.
        $\circ$ Set $t := t + 1$.
        $\circ$ **if** $v_k(s_t, a_t) = N_{t_k}(s_t, a_t)$ **then** terminate current episode.
        $\circ$ **if**

$$(t - t_k + 1)\widetilde{\rho}_k - \sum_{\tau=t_k}^{t} r_\tau > \Gamma_t(\bar{D}) \tag{9}$$

    **then** set $\Phi := \Phi \setminus \{\phi_k\}$ and terminate current episode.
**end for**

---

The UCB-MS algorithm we propose (shown as Alg. 1) basically performs the policy computation of UCRL2 for each model $\phi$. That is, in episodes $k = 1, 2, \ldots$, UCB-MS constructs for each state representation $\phi \in \Phi$ a set of plausible MDPs $\mathcal{M}_{k,\phi}$ and computes the optimistic average reward

$$\widetilde{\rho}_{k,\phi} = \operatorname*{argmax}_{\pi \in \Pi^{\mathrm{SD}}, M \in \mathcal{M}_{k,\phi}} \{\rho(\pi, M)\}. \tag{6}$$

This problem can be solved using Extended Value Iteration (EVI) [9] up to an arbitrary accuracy.[3] Among all the models, UCB-MS selects the one with highest average reward $\widetilde{\rho}_k$, cf. Eq. (8). The associated optimistic policy $\widetilde{\pi}_k$ is executed until the number of visits is doubled in at least one state-action pair (UCRL2 stopping condition) or this policy does not provide sufficiently high average reward (see Eq. 9), in which case the model $\phi_k$ is eliminated.

The function $\Gamma_t$ in Eq. (9) defines the allowed deviation from the promised optimistic average reward $\widetilde{\rho}_k$. We define $\Gamma_t$ according to Lemma 3 as

$$\Gamma_t(D) := \left( 2D\sqrt{14 S_{\phi_t} \log\left(\frac{4At_{k(t)}^3}{\delta}\right)} + 2 \right) \sum_{s,a} \frac{v_{k(t)}(s,a)}{\sqrt{N_{k(t)}(s,a)}} + 4D\sqrt{T_{k(t)} \log\left(\frac{16t_{k(t)}^2 T_{k(t)}}{\delta}\right)} + D, \tag{7}$$

where $k(t)$ denotes the episode in which step $t$ occurs. In Eq. 9 we exploit the prior knowledge $\bar{D} \geq D$ in order to properly define the condition for model elimination. We will see below in Section 5 that it is easy to adapt the algorithm to the case of unknown diameter.

If the set $\Phi$ consists only of a single Markov model, UCB-MS basically coincides with UCRL with an additional checking step that will result in discarding the single model only with small probability. Note that UCB-MS shares the optimistic model selection and the idea of eliminating underachieving models with OMS, however its structure is much simpler.

Concerning the computational complexity of our algorithm, note that the EVI subroutine we use for policy computation works just as ordinary value iteration with the same convergence properties and the same computational complexity with an additional overhead of $O(S^2 A)$ per iteration step, cf. [9]. Further, policy computation is performed for each model $\phi$ at most $|\Phi| + S_\phi A \log T$ times, cf. Lemma 5 (c) below.

Our first result is the following regret bound for UCB-MS. Here $S_{\max} := \max_\phi S_\phi$ denotes the size of state space of the largest model and $S_\Sigma := \sum_\phi S_\phi$ the size of the total state space over all models.

**Theorem 4.** *With probability $1 - \delta$, the regret of* UCB-MS *using $\bar{D} \geq D$ is bounded by*

$$const \cdot \bar{D}\sqrt{S_{\max}S_\Sigma AT}\log\left(\tfrac{T}{\delta}\right).$$

Note that the bound of Theorem 4 holds for *any* Markov model in $\Phi$. Thus, in case there is a Markov model with smaller state space the regret bound shows that UCB-MS automatically adapts to this preferable model. When $\Phi$ consists of a single Markov model we re-establish the bounds for UCRL2 (however for an algorithm that unlike UCRL2 needs the diameter $D$ as input). Most importantly, the bound of Theorem 4 improves over the currently best known bound for BLB, which is of order $\widetilde{O}(T^{2/3})$. If all models induce a state space of equal size $S$, the bound in Theorem 4 is $\widetilde{O}(DS\sqrt{|\Phi|AT})$, which also improves over the claimed regret bound of OMS, which is of order $\widetilde{O}(DS^{3/2}A\sqrt{|\Phi|T})$. We note however that in other cases the state space dependence of the OMS bound may be better. In Section 6 below we show how to regain the OMS bound for our algorithm and how $S_\Sigma$ in the bounds can be replaced by the *effective* size of the state space, which in some cases (like for hierarchical models) can be considerably smaller.

While the $\sqrt{A}$-dependence is optimal as for UCRL2, by using a refined analysis (see [13]) it is also possible to obtain an optimal $\sqrt{D}$-dependence. On the other hand, the optimality in $S$ and $|\Phi|$ is still an open question. While the $S$-dependence can be reduced using Bernstein inequality, we are not aware of any lower bound for $|\Phi|$ in this setting. The closest result we know is for aggregation techniques with *full* information where it is possible to obtain bounds of order $\log(|\Phi|)$. Obviously, in our setting we have less information and it is not clear if it is possible to obtain logarithmic dependence.

Note that while the regret is measured w.r.t. the true Markov model $\phi^\circ$, it is actually not necessary to identify $\phi^\circ$ to obtain the regret bound of Theorem 4. As long as a non-Markov model gives at least the same reward that would be expected from a Markov model there is no need to discard it. Such a model could be, for example, a good (non-Markovian) approximation.

### 4.1 Analysis (Proof of Theorem 4)

The following lemma collects some basic facts about UCB-MS.

**Lemma 5.** *With probability $1 - \delta$, all of the following statements hold:*

*(a) The confidence intervals* (1) *and* (2) *of the Markov model $\phi^\circ$ hold for all time steps $t = 1, \ldots, T$.*

*(b) No Markov models are discarded in* (9).

*(c) The number of episodes of* UCB-MS *is bounded by $|\Phi| + S_\Sigma A \log T$.*

*Proof.* (a) follows from Lemma 1 by summing over the error probabilities giving a total error probability of $\sum_t \frac{\delta}{30t^8} < \frac{\delta}{6}$.

For (b), if UCB-MS chooses a Markov model, then the regret in the respective episode is bounded according to Lemma 3. The sum over the respective error probabilities $\delta/2t_k^2$ over all episodes is bounded by $\frac{5\delta}{6}$, which proves (b).

If (b) holds, then there are at most $|\Phi| - 1$ episodes in which a model is discarded. For episodes which are terminated by doubling the number of visits, we can use Proposition 18 of [9], as the episode termination criterion of UCB-MS is the same as for UCRL2. Since we have to take into account all states of all models, the size of the state space to be considered is the sum over the sizes of the state spaces of the individual models. $\qquad\square$

The bound on the number of episodes in the worst case depends on $S_\Sigma$. Under some assumptions on the given models in $\Phi$ (like having hierarchical models) this can be reduced, see Section 6 for details.

*Proof of Theorem 4.* We assume that the statements of Lemma 5 all hold, which is the case with probability $1 - \delta$. Let $\phi^\circ$ be a Markov model in $\Phi$ and consider any episode $k$. By Lemma 5 (a),

the optimistic estimate $\widetilde{\rho}_{k,\phi^\circ} \geq \rho^*(\phi^\circ)$. By the optimism of the algorithm we further have that $\widetilde{\rho}_k \geq \widetilde{\rho}_{k,\phi^\circ}$. Hence, the regret $\Delta_k$ in each episode $k$ is bounded by

$$\Delta_k := T_k \cdot \rho^*(\phi^\circ) - \sum_{\tau=t_k}^{t_k+T_k} r_\tau \leq T_k \cdot \widetilde{\rho}_k - \sum_{\tau=t_k}^{t_k+T_k} r_\tau.$$

By the definition of the algorithm, condition (9) does not hold at least up to the final step of the episode, so that we obtain that (as rewards are upper bounded by 1)

$$\Delta_k \leq \Gamma_{t_k}(\bar{D}) + 1.$$

Using the definition of $\Gamma_t(\bar{D})$ in (7) and summing over all $K$ episodes, we obtain a regret bound of

$$\sum_k \Delta_k \leq \sum_k (\Gamma_{t_k}(\bar{D}) + 1)$$

$$\leq \left(2\bar{D}\sqrt{14 S_{\max} \log\left(\frac{4AT^3}{\delta}\right)} + 2\right) \sum_k \sum_{s,a} \frac{v_k(s,a)}{\sqrt{N_k(s,a)}} + 4\bar{D}\sqrt{\log\left(\frac{16T^3}{\delta}\right)} \sum_k \sqrt{T_k} + K\bar{D}.$$

Using that $\sum_k T_k = T$ together with Jensen's inequality, we have $\sum_k \sqrt{T_k} \leq \sqrt{KT}$. Further, as for the analysis of UCRL2, we have that (cf. Eq. 20 of [9])

$$\sum_k \sum_{s,a} \frac{v_k(s,a)}{\sqrt{N_k(s,a)}} \leq (\sqrt{2}+1)\sqrt{S_\Sigma AT}.$$

Summarizing, we obtain using the bounds on the number of episodes of Lemma 5 (c) and noting that $|\Phi| \leq S_\Sigma$ after some simplifications a regret bound of

$$const_1 \cdot \bar{D}\sqrt{S_{\max} S_\Sigma AT \log\left(\frac{T}{\delta}\right)} + const_2 \cdot \bar{D}\sqrt{S_\Sigma AT (\log T)\left(\log \frac{T}{\delta}\right)} + const_3 \cdot \bar{D} S_\Sigma A \log T,$$

which completes the proof of the theorem. □

## 5  Unknown Diameter

If the diameter is unknown we suggest the following guessing scheme. We run UCB-MS with some initial value $\bar{D} \geq 1$. If at some step *all models have been eliminated* then double the value of $\bar{D}$ and restart the algorithm, that is, start a new episode now *considering all models again*.

One can show that the regret of this doubling scheme is basically bounded as before unless $D$ is very large compared to $T$.

**Theorem 6.** *With probability* $1 - \delta$*, the regret of* UCB-MS *guessing* $D$ *by doubling is bounded by*

$$const \cdot D\sqrt{\left(S_{\max} S_\Sigma A + |\Phi| \log D\right) T \log\left(\frac{T}{\delta}\right)}.$$

*Proof.* Let $D_k$ denote the parameter $\bar{D}$ used in episode $k$ as an estimate for $D$. As in the proof of Theorem 4 we have that a Markov model will not be eliminated with high probability once $D_k \geq D$. Hence, in total there cannot be more than $\lceil |\Phi| \log_2 D \rceil$ episodes that are terminated by discarding a model.

Let $\Gamma_t(D)$ be defined as in (7). Then the same argument as in the proof of Theorem 4 shows that the regret in each episode $k$ is bounded by $\Gamma_{t_k}(D_k) + 1$.

The rest of the proof can be rewritten from Theorem 4 using that $D_k < 2D$ for all $k$ with high probability. The only difference is that the bound on the number of episodes has an additional term of $\lceil |\Phi| \log_2 D \rceil$, so that one obtains a regret bound of

$$const_1 \cdot D\sqrt{S_{\max} S_\Sigma AT \log\left(\frac{T}{\delta}\right)} + const_2 \cdot D\sqrt{\left(S_\Sigma A(\log T) + |\Phi| \log D\right) T \log \frac{T}{\delta}} +$$

$$const_3 \cdot \left(D S_\Sigma A + |\Phi| \log D\right) \log T.$$

Summarizing the terms gives the claimed bound. □

Theorem 6 shows that the cost of the guessing scheme w.r.t. the regret is small and, in particular, does not result in any additive constant in the bound that is exponential in the diameter (in contrast to the bound for BLB [4]). Thus, the improvements over OMS discussed after Theorem 4 hold also for UCB-MS with guessing the diameter.

## 6 Improving the Bounds

In this section, we consider further improvements of our bounds and introduce the notion of the *effective* size of the state space for a set of models $\Phi$.

### 6.1 Improving on the Number of Episodes

The regret bounds we obtain for UCB-MS are basically of the same order as for standard reinforcement learning in MDPs (i.e. with a given Markov model) as achieved e.g. by [9]. However, the state space dependence seems not completely satisfactory, as the bounds do not only depend on the state space size of the Markov model, but on the total state space size $S_\Sigma$ over all models.

The appearance of the parameter $S_\Sigma$ in the bounds is due to the bound on the number of episodes in Lemma 5 (c). In the worst case, this bound cannot be improved. That is, without any further assumptions on the way models in $\Phi$ aggregate histories one cannot say how visits in a state under some model $\phi$ translate into state visits under some other model $\phi'$. For example, when under some model $\phi$ all states have been visited so far, the respective histories may be mapped to just a single state under some other model $\phi'$. Consequently, one basically has to assume that the states of different models $\phi, \phi'$ are completely independent of each other, which leads to the bound of Lemma 5 (c).

However, if there is some particular structure on the set of given models $\Phi$, the bound on the number of episodes can be improved to not depend on the total number of states $S_\Sigma$.

**Definition 7.** *Let $\Phi$ be a set of state representation models. We define the effective size $S_\Phi$ of the state space of $\Phi$ to be the number of states that are sufficient to cover all states under $\Phi$ in the sense that visits in all $S_\Phi$ covering states induce visits in all other states.*

A simple example is when models are hierarchical. That is, there is some model $\phi$ in $\Phi$, such that all other models $\phi'$ aggregate the states of $\phi$, i.e., it holds that if $\phi(h) = \phi(h')$ then $\phi'(h) = \phi'(h')$ for all histories $h, h'$ in $\mathcal{H}$. In this case, $S_\Phi = S_\phi$, while $S_\Sigma$ could be of order $2^{S_\phi}$, as each subset of $\mathcal{S}_\phi$ may correspond to an aggregated state in some other model of $\Phi$. Note that when considering different orders for an MDP, this also results in a hierarchical model set.

In general, we obviously have that $S_\Phi \leq S_\Sigma$ and the bound on the number of episodes of Lemma 5 (c) can be improved to depend on $S_\Phi$ instead of $S_\Sigma$ (with the same proof).

**Lemma 8.** *The number of episodes of UCB-MS terminated by the doubling criterion is bounded by $S_\Phi A \log T$.*

Accordingly, we can strengthen the results of Theorems 4 and 6 as follows.

**Theorem 9.** *The regret bounds of Theorems 4 and 6 hold with $S_\Sigma$ replaced by $S_\Phi$.*

### 6.2 Improving Further on the State Space Dependence

Even after replacing $S_\Sigma$ by $S_\Phi$, there is still room for improvement of the bounds with respect to the size of the state space. In principle, one would like to have a dependence on the size of the state space of the Markov model $\phi^\circ$. As we have seen, with the current analysis the dependence on the effective number of states $S_\Phi$ is unavoidable. However, we can improve over the second appearing state space term $S_{\max}$ by guessing the right size of the state space (i.e., $S_{\phi^\circ}$). We distinguish between two cases, depending on whether a bound on the diameter is known.

#### 6.2.1 Diameter Known

If there is a known bound on the diameter, we can guess the size of the state space by the same scheme we have suggested for guessing the diameter in Section 5. That is, starting with $S := 1$ or $S := \min_\phi S_\phi$ we compare the collected rewards to the optimistic average reward $\widetilde{\rho}_k$ of the current episode $k$, as before eliminating underachieving models. As comparison term we choose now (in accordance with the regret bound for UCRL2 in Theorem 2)

$$\Gamma_t(S) := 34 D S \sqrt{A(t - t_k + 1) \log\left(\tfrac{2t^3}{\delta}\right)}. \tag{10}$$

For this guessing scheme one can show the following regret bound.

**Theorem 10.** *With probability $1 - \delta$, the regret of UCB-MS guessing $S$ by doubling is bounded by*

$$const \cdot D S_{\phi^\circ} \sqrt{\left(S_\Phi A \log T + |\Phi| \log S_{\phi^\circ}\right) AT \log\left(\tfrac{T}{\delta}\right)}.$$

*Proof.* The proof is like that for Theorem 6 only that now $S$ instead of $D$ is guessed and the comparison term $\Gamma_t$ is different. That is, any Markov model $\phi^\circ$ will not be discarded with high probability once $S \geq S_{\phi^\circ}$. Therefore, there will be at most $\lceil |\Phi| \log_2 S_{\phi^\circ} \rceil$ episodes that are terminated by discarding a model.

Let $S_k$ be the guess for the size of the state space in episode $k$. Then, similar to the proofs of Theorems 4 and 6, the regret in each episode $k$ is bounded by $\Gamma_{t_k}(S_k) + 1$. As $S_k \leq 2 S_{\phi^\circ}$ w.h.p., summing over all $\leq \lceil |\Phi| \log_2 S_{\phi^\circ} \rceil + S_\Phi A \log T$ episodes, Jensen's inequality gives the claimed regret bound. $\qquad\square$

We see that replacing $S_{\max}$ with $S_{\phi^\circ}$ comes at a cost of worse dependence on the number of states and actions, as the summing over episodes in the proof has to be done differently. Still, if $S_{\max}$ is quite large, the bound of Theorem 10 can be an improvement over the previously presented bounds.

### 6.2.2 Unknown Diameter

If the diameter is not known, one can do the guessing for both $D$ and $S$ at the same time. More precisely, in the comparison term one does not guess $D$ and $S$ separately but the factor $DS$ instead. That is, one starts with setting $\widetilde{DS} := 1$ or some other fixed value like $\widetilde{DS} := \min_\phi S_\phi$ and defines the comparison term as

$$\Gamma_t(\widetilde{DS}) := 34 \widetilde{DS} \sqrt{A(t - t_k + 1) \log\left(\frac{2t^3}{\delta}\right)}. \tag{11}$$

This leads to the following regret bound, which basically corresponds to the bound that has been claimed for OMS, only with $S_\Sigma$ replaced by the potentially smaller $S_\Phi$.

**Theorem 11.** *With probability $1 - \delta$, the regret of* UCB-MS *guessing both $D$ and $S$ by doubling is bounded by*

$$const \cdot DS_{\phi^\circ} \sqrt{\left(S_\Phi A \log T + |\Phi| \log(DS_{\phi^\circ})\right) AT \log\left(\frac{T}{\delta}\right)}.$$

*Proof.* The proof is like that for Theorem 10. W.h.p. there will be at most $\lceil |\Phi| \log_2(DS_{\phi^\circ}) \rceil$ episodes that are terminated by eliminating a model, while the regret in each episode $k$ is bounded by $\Gamma_{t_k}(\widetilde{DS}_k) + 1$, where $\widetilde{DS}_k \leq 2 DS_{\phi^\circ}$ is the guess for episode $k$. A sum over the episodes gives the claimed bound. $\qquad\square$

## 7 Discussion

While we have decided to use UCRL2 as reference algorithm for the definition of our UCB-MS strategy, our approach can actually serve as a blueprint for adapting any optimistic algorithm with known regret bounds to the state representation setting considered in this paper. In particular, improved regret bounds (possible w.r.t. the parameters $S$ and $D$, cf. [9]) for UCRL2 or a variation of it (such as the recent [13]) automatically result in improved bounds for a corresponding variant of UCB-MS.

The OMS algorithm [5] employs some form of regularization so that models with large state space are less appealing. However, this did not avoid the dependence of the claimed bounds of [5] on $S_\Sigma$. It is an interesting question whether some improved regularization approach can give bounds that only depend on $S_{\phi^\circ}$. In general, the right dependence of regret bounds on the size of the model set $\Phi$ is also an open problem.

Another question that is still open also for the MDP setting is whether the diameter can be replaced by the *bias span* $\lambda^*$ of the optimal policy [10, 14]. With an upper bound on $\lambda^*$, one could replace UCRL2 by the SCAL algorithm of [14]. However, the guessing scheme we employ for the diameter does not work for SCAL, as chosen policies may not be optimistic anymore, if the guess for $\lambda^*$ is too small.

Another direction for future research are generalizations to infinite model sets, which for the case of discrete sets has already been done for the BLB algorithm [6]. Parametric sets of models would be an interesting next step from there. In this context, it also makes sense to consider *approximate* Markov models, that is, the assumption that there is a Markov model is dropped. The results given in [15] for this setting are also affected by the mentioned error in the proof of the OMS regret bound. We think that our approach can be adapted, however the details still have to be worked out.

**Acknowledgments**

This work has been supported by the Austrian Science Fund (FWF): I 3437-N33 in the framework of the CHIST-ERA ERA-NET (DELTA project). Odalric-Ambrym Maillard was supported by CPER Nord-Pas de Calais/FEDER DATA Advanced data science and technologies 2015-2020, the French Ministry of Higher Education and Research, Inria Lille – Nord Europe, CRIStAL, and the French Agence Nationale de la Recherche, under grant ANR-16-CE40-0002 (project BADASS).

## Footnotes

[1] The confidence intervals shown here are the ones we use in the following and slightly differ from the confidence intervals given for UCRL2 in [9]. That is, the confidence $\delta$ of the original values is replaced by $\delta/2t^2$ to guarantee smaller error probability, which is needed in our analysis.

[2] As noted before, the error probability $\delta$ has been changed from $\delta$ to $\delta/2t^2$ here.

[3] As for UCRL2, we set the accuracy in episode $k$ to be $1/\sqrt{t_k}$.

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
