[Supplementary Material · LearningStateRepresentations.pdf]

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

 that depend on $S_\Phi$, which gives improved bounds, e.g. for hierarchical models.

**Overview.** We start with describing the learning setting in full detail in the following section. In Section 3, we give some preliminaries concerning the UCRL2 algorithm. Our UCB-MS algorithm is introduced in Section 4 where we also give the regret analysis in case the diameter of the underlying Markov model is known. The following Section 5 shows how to guess the diameter otherwise. Section 6 gives some further improvements and also introduces the notion of effective state space.

## 2   Setting

The details of the considered learning setting are as follows. At each time step $t = 1, 2, \ldots$, the learner receives an initial observation $o_t$ and has to choose an action $a_t$ from a finite set of actions $\mathcal{A}$. In return, the learner receives a reward $r_t$ taken from $\mathcal{R} = [0, 1]$ and the next observation $o_{t+1}$.

We denote by $\mathcal{O}$ the set of observations and define the history $h_t$ up to step $t$ as the sequence $o_1, a_1, r_1, o_2, \ldots, a_t, r_t, o_{t+1}$ of observations, actions and rewards. The set $\mathcal{H}_t := \mathcal{O} \times (\mathcal{A} \times \mathcal{R} \times \mathcal{O})^{t-1}$ contains all possible histories up to step $t$ and we set $\mathcal{H} := \bigcup_{t \geq 1} \mathcal{H}_t$ to be the set of all possible histories.

### 2.1   Models and MDPs

A *state-representation model* (in the following short: *model*) $\phi$ is a function that maps histories to states, that is, $\phi : \mathcal{H} \to \mathcal{S}_\phi$. If a model $\phi$ induces a *Markov decision process (MDP)* we call it a *Markov model*. An MDP has the Markov property that any time $t$, the probability of reward $r_t$ and next state $s_{t+1} = \phi(h_{t+1})$, given the past history $h_t$, only depends on the current state $s_t = \phi(h_t)$

89  and the chosen action $a_t$, i.e., $P(s_{t+1}, r_t | h_t, a_t) = P(s_{t+1}, r_t | s_t, a_t)$. We assume that for MDPs this
90  probability is also independent of $t$.

91  Usually an MDP $M$ is denoted as a tuple $M = (\mathcal{S}_\phi, \mathcal{A}, r, p)$, where $r(s, a)$ is the mean reward and
92  $p(s'|s, a)$ the probability of a transition to state $s' \in \mathcal{S}_\phi$ when choosing action $a \in \mathcal{A}$ in state $s \in \mathcal{S}_\phi$.
93  If $\phi$ is a Markov model, we write the induced MDP as $M(\phi)$.

94  MDPs are called *communicating* if for any two states $s, s'$ it is possible to reach $s'$ from $s$ with
95  positive probability by choosing appropriate actions. The smallest expected time it takes to connect
96  any two states is called the *diameter $D$* of the MDP, cf. [9]. In communicating MDPs, the optimal
97  average reward $\rho^*$ is independent of the initial state and will be achieved by a stationary deterministic
98  policy $\pi^* \in \Pi^{\text{SD}}$ that maps states to actions. For a Markov model $\phi$, the diameter and the optimal
99  average reward of the induced MDP will be denoted as $D(\phi)$ and $\rho^*(\phi)$, respectively.

## 2.2 Problem setting

101  The learning setting we consider is the following. As already described before, the learner chooses
102  actions $a_t$ and obtains a reward $r_t$ and an observation $o_{t+1}$ in return. We assume that the learner has
103  a finite set $\Phi$ of models at her disposal and at least one model $\phi^\circ$ in $\Phi$ is a Markov model. The goal is
104  to provide algorithms that perform well with respect to the optimal policy $\pi^*$ in the MDP $M(\phi^\circ)$,
105  that is, the optimal strategy when the Markov model and the induced underlying MDP are completely
106  known. Accordingly, the performance of a learning algorithm will be measured by considering its
107  *regret* after any $T$ steps defined as (cf. [9, 10, 4])

$$T\rho^*(\phi^\circ) - \sum_{t=1}^{T} r_t\,,$$

108  where $r_t$ is the reward received by the learning algorithm at step $t$.

# 3 UCRL2 Preliminaries

110  The algorithm we propose is based on the UCRL2 algorithm of [9]. Thus, in this section we give
111  some preliminaries concerning the UCRL2 algorithm.

112  UCRL2 is an algorithm that generalizes the *optimism in the face of uncertainty* idea of UCB [12]
113  from the bandit setting to reinforcement learning in MDPs. The algorithm maintains estimates of
114  rewards and transition probabilities and respective confidence intervals that make up a set of plausible
115  MDPs $\mathcal{M}$.

116  That is, acting in an unknown MDP, UCRL2 maintains estimates $\hat{r}(s, a)$ and $\hat{p}(\cdot|s, a)$ of rewards and
117  transition probabilities, respectively. The set $\mathcal{M}_t$ of plausible MDPs at step $t$ contains all MDPs with
118  rewards $\widetilde{r}(s, a)$ and $\widetilde{p}(\cdot|s, a)$ and transition probabilities satisfying[1]

$$\left| \hat{r}(s, a) - \widetilde{r}(s, a) \right| \leq \sqrt{\tfrac{7 \log(4SAt^3/\delta)}{2N(s,a)}}, \tag{1}$$

$$\left\| \hat{p}(\cdot|s, a) - \widetilde{p}(\cdot|s, a) \right\|_1 \leq \sqrt{\tfrac{14S \log(4At^3/\delta)}{2N(s,a)}}, \tag{2}$$

119  where $N(s, a)$ denotes the number of times $a$ has been chosen in $s$ (and is set to 1, if $a$ has not been
120  chosen in $s$ so far). The true MDP $M$ is in $\mathcal{M}$ with high probability.

121  **Lemma 1** (Lemma 17 in the appendix of [9][2]). *With probability at least $1 - \frac{\delta}{30t^8}$, at step $t$ the true*
122  *MDP $M$ is contained in the set $\mathcal{M}_t$.*

123  The UCRL2 algorithm proceeds in episodes $k = 1, 2, \ldots$. In each episode $k$ starting at step $t_k$
124  the algorithm plays a fixed policy $\widetilde{\pi}_k$, which is chosen to maximize the optimal average reward in
125  $\mathcal{M}_k := \mathcal{M}_{t_k}$. That is, writing $\rho(\pi, M)$ for the average reward of policy $\pi$ in MDP $M$ we have
126  $\widetilde{\rho}_k := \max_{\pi, M \in \mathcal{M}_k} \rho(\pi, M) = \rho(\widetilde{\pi}_k, \widetilde{M}_k)$, where $\widetilde{M}_k$ is an optimistic MDP chosen from $\mathcal{M}_k$ to
127  maximize $\widetilde{\rho}_k$. As the true MDP $M$ is in $\mathcal{M}_k$ with high probability, we also have that $\widetilde{\rho}_k \geq \rho^*$.

Let $v_k(s, a)$ denote the number of times $a$ has been chosen in $s$ in episode $k$, while $N_k(s, a)$ denotes the number of times $a$ has been chosen in $s$ before episode $k$ (i.e., in episodes 1 to $k - 1$). If there were no visits in $(s, a)$ before episode $k$, then $N_k(s, a)$ is set to 1. Episode $k$ is terminated by UCRL2 when a state $s$ is reached in which $v_k(s, \widetilde{\pi}_k(s)) = N_k(s, \widetilde{\pi}_k(s))$.

Let $S = |\mathcal{S}|$ be the size of the state space, $A = |\mathcal{A}|$ the size of the action space, and $D$ be the diameter of the MDP. Then, one can show the following upper bound on the regret of UCRL2.

**Theorem 2** (Theorem 2 of [9]). *With probability $1 - \delta$ the regret of UCRL2 after any $T$ steps is bounded by*

$$34DS\sqrt{AT \log\left(\frac{2T^3}{\delta}\right)}.$$

The bound is based on an episode-wise decomposition of the regret, which we will use for our algorithm. Let $T_k$ be the (current) length of episode $k$. In the following, we abuse notation for $T_k$ as well as for $v_k(s, a)$ by using the same notation for the number of steps in a terminated episode as well as for the current number of steps in an ongoing episode. Further, recall that $t_k$ denotes the time step when episode $k$ starts. The regret of UCRL2 in any episode $k$ is bounded as follows.[3]

**Lemma 3.** *Consider an arbitrary episode $k$ started at step $t_k$. With probability $1 - \frac{\delta}{2t_k^2}$, the regret of UCRL2 at each step $T_k$ in this episode is bounded by*

$$\left(2D\sqrt{14S \log\left(\frac{16t_k^3}{\delta}\right)} + 2\right) \sum_{s,a} \frac{v_k(s,a)}{\sqrt{N_k(s,a)}} + 2D\sqrt{5T_k \log\left(\frac{16t_k^2 T_k}{\delta}\right)} + D.$$

## 4 The UCB-MS Algorithm

Now let us turn to the state representation learning setting introduced in Section 2. We start with the simpler case when an upper bound $\bar{D}$ on the diameter $D := D(\phi^\circ)$ of the Markov model $\phi^\circ$ is known (i.e., $\bar{D} \geq D$). The case when no bound on the diameter is known is dealt with in Section 5.

The UCB-MS algorithm we propose (shown as Alg. 1) basically performs the policy computation of UCRL2 for each model $\phi$. That is, in episodes $k = 1, 2, \ldots$, UCB-MS constructs for each state representation $\phi \in \Phi$ a set of plausible MDPs $\mathcal{M}_{k,\phi}$ and computes the optimistic average reward

$$\widetilde{\rho}_{k,\phi} = \operatorname*{argmax}_{\pi \in \Pi^{\mathrm{SD}}, M \in \mathcal{M}_{k,\phi}} \{\rho(\pi, M)\}. \tag{3}$$

This problem can be solved using Extended Value Iteration (EVI) [9] up to an arbitrary accuracy.[4] Among all the models, UCB-MS selects the one with highest average reward (i.e., $\phi_k := \operatorname{argmax}_{\phi \in \Phi}\{\widetilde{\rho}_{k,\phi}\}$). The associated optimistic policy $\widetilde{\pi}_{k,\phi_k}$ is executed until the number of visits is doubled in at least one state-action pair (UCRL2 stopping condition) or this policy does not provide sufficiently high average reward (see Eq. 6), in which case the model $\phi_k$ is eliminated.

The function $\Gamma_t$ in Eq. (6) defines the allowed deviation from the promised optimistic average reward $\widetilde{\rho}_k := \widetilde{\rho}_{k,\phi_k}$. We define $\Gamma_t$, according to Lemma 3, as

$$\Gamma_t(D) := \left(2D\sqrt{14S_{\phi_t} \log\left(\frac{16t_{k(t)}^3}{\delta}\right)} + 2\right) \sum_{s,a} \frac{v_{k(t)}(s,a)}{\sqrt{N_{k(t)}(s,a)}} + 2D\sqrt{5T_{k(t)} \log\left(\frac{16t_{k(t)}^2 T_{k(t)}}{\delta}\right)} + D,$$

$$\tag{4}$$

where $k(t)$ denotes the episode in which step $t$ occurs. In Eq. 6 we exploit the prior knowledge $\bar{D} \geq D$ in order to properly define the condition for model elimination. We will see below in Section 5 that it is easy to adapt the algorithm to the case of unknown diameter.

If the set $\Phi$ consists only of a single Markov model, basically UCB-MS coincides with UCRL with an additional checking step that will result in discarding the single model only with small probability. Note that UCB-MS shares the optimistic model selection and the idea of eliminating underachieving models with OMS, however its structure is much simpler.

Concerning the computational complexity of our algorithm, note that the EVI subroutine we use for policy computation works just as ordinary value iteration with the same convergence properties

**Algorithm 1** UCB-Model Selection (UCB-MS)

---

**Input:** set of models $\Phi$, confidence parameter $\delta \in (0,1)$, upper bound $\bar{D}$ on diameter
**Initialization:** Let $t := 1$ be the current time step.
**for** episodes $k = 1, 2, \ldots$ **do**

    Let $t_k := t$ be the initial step of the current episode $k$.

    $\triangleright$ For each $\phi \in \Phi$, use Extended Value Iteration (EVI) to compute an optimistic MDP $\widetilde{M}_k(\phi)$ in $\mathcal{M}_{t,\phi}$ (the set of *plausible* MDPs defined via the confidence intervals (1) and (2) for the estimates so far), a (near-)optimal policy $\widetilde{\pi}_{k,\phi}$ on $\widetilde{M}_{t,\phi}$ with approximate average reward $\widetilde{\rho}_{t,\phi}$.

    $\triangleright$ Choose model $\phi_k \in \Phi$ such that

$$\phi_k = \underset{\phi \in \Phi}{\operatorname{argmax}} \left\{ \widetilde{\rho}_{t,\phi} \right\}, \tag{5}$$

    and set $\widetilde{\rho}_k := \rho_{t,\phi_k}$, $\widetilde{\pi}_k := \widetilde{\pi}_{t,\phi_k}$, and $\mathcal{S}_k := \mathcal{S}_{\phi_k}$.

    $\triangleright$ Repeat till termination of the current episode $k$:
        $\circ$ Choose action $a_t := \pi_k(s_t)$, get reward $r_t$ and observe next state $s_{t+1} \in \mathcal{S}_k$
        $\circ$ Set $t := t + 1$.
        $\circ$ **if** $v_k(s_t, a_t) = N_{t_k}(s_t, a_t)$ **then** terminate current episode.
        $\circ$ **if**

$$(t - t_k + 1)\widetilde{\rho}_k - \sum_{\tau = t_k}^{t} r_\tau > \Gamma_t(\bar{D}) \tag{6}$$

    **then** set $\Phi := \Phi \setminus \{\phi_k\}$ and terminate current episode.
**end for**

---

and the same computational complexity with an additional overhead of $O(|S|^2|A|)$ per iteration step, cf. [9]. Policy is computed for each model $\phi$ at most $|\Phi| + S_\phi A \log T$ times, cf. Lemma 5 (c) below.

Our first result is the following regret bound for UCB-MS. Here $S_{\max} := \max_\phi S_\phi$ denotes the size of state space of the largest model and $S_\Sigma := \sum_\phi S_\phi$ the size of the total state space over all models.

**Theorem 4.** *With probability* $1 - \delta$, *the regret of* UCB-MS *using* $\bar{D} \geq D$ *is bounded by*

$$const \cdot \bar{D} \sqrt{S_{\max} S_\Sigma A T} \log\left(\tfrac{T}{\delta}\right).$$

Note that the bound of Theorem 4 holds for any Markov model in $\Phi$. Thus, in case there is a Markov model with smaller state space the regret bound shows that UCB-MS automatically adapts to this preferable model. When $\Phi$ consists of a single Markov model we re-establish the bounds for UCRL2 (up to the prior knowledge). Most importantly, the bound of Theorem 4 improves over the currently best known bound for BLB, which is of order $\widetilde{O}(T^{2/3})$. If all models induce a state space of equal size $S$, the bound in Theorem 4 is $\widetilde{O}(DS\sqrt{|\Phi|AT})$, which also improves over the claimed regret bound of OMS, which is of order $\widetilde{O}(DS^{3/2}A\sqrt{|\Phi|T})$. We note however that in other cases the state space dependence of the OMS bound may be better. In Section 6 below we show how to regain the OMS bound for our algorithm and how in some cases (like for hierarchical models) the dependence on $S_\Sigma$ can be replaced by the smaller *effective* size of the state space.

## 4.1 Analysis (Proof of Theorem 4)

The following lemma collects some basic facts about UCB-MS.

**Lemma 5.** *With probability* $1 - \delta$, *all of the following statements hold:*

*(a) The confidence intervals (1) and (2) of the Markov model $\phi^\circ$ hold for all time steps $t = 1, \ldots, T$.*

*(b) No Markov models are discarded in (6).*

*(c) The number of episodes of* UCB-MS *is bounded by $|\Phi| + S_\Sigma A \log T$.*

*Proof.* (a) follows from Lemma 1 by summing over the error probabilities giving an error probability of $\sum_t \frac{\delta}{30t^8} < \frac{\delta}{6}$.

For (b), if UCB-MS chooses a Markov model, then the regret in the respective episode is bounded according to Lemma 3. The sum over the respective error probabilities $\delta/2t_k^2$ over all episodes is bounded by $\frac{5\delta}{6}$, which proves (b).

If (b) holds, then there are at most $|\Phi| - 1$ episodes in which a model is discarded. For episodes which are terminated by doubling the number of visits, we can use Proposition 18 of [9], as the episode termination criterion of UCB-MS is the same as for UCRL2. Since we have to take into account all states of all models, the size of the state space to be considered is the sum over the sizes of the state spaces of the individual models. $\qquad\square$

The bound on the number of episodes in the worst case depends on $S_\Sigma$. Under some assumptions on the given models in $\Phi$ (like having hierarchical models) this can be reduced, see Section 6 for details.

*Proof of Theorem 4.* We assume that the statements of Lemma 5 all hold, which is the case with probability $1 - \delta$. Let $\phi^\circ$ be a Markov model in $\Phi$ and consider any episode $k$. By Lemma 5 (a), the optimistic estimate $\widetilde{\rho}_{t_k,\phi^\circ} \geq \rho^*(\phi^\circ)$. By the optimism of the algorithm we further have that $\widetilde{\rho}_k \geq \widetilde{\rho}_{t_k,\phi^\circ}$. Hence, the regret $\Delta_k$ in each episode $k$ is bounded by

$$\Delta_k := T_k \cdot \rho^*(\phi^\circ) - \sum_{\tau=t_k}^{t_k+T_k} r_\tau \ \leq\ T_k \cdot \rho_k - \sum_{\tau=t_k}^{t_k+T_k} r_\tau.$$

By the definition of the algorithm, condition (6) does not hold at least up to the final step of the episode, so that we obtain that (as rewards are upper bounded by 1)

$$\Delta_k \leq \Gamma_{t_k}(\bar{D}) + 1.$$

Using the definition of $\Gamma_t(\bar{D})$ (see (4)) and writing $K$ for the total number of episodes, we obtain for the total regret summing over all episodes a bound of

$$\sum_k \Delta_k \ \leq\ \sum_k \left(\Gamma_{t_k}(\bar{D}) + 1\right)$$

$$\leq \left(2\bar{D}\sqrt{14 S_{\max} \log\left(\tfrac{16T^3}{\delta}\right)} + 2\right)\sum_k \sum_{s,a} \tfrac{v_k(s,a)}{\sqrt{N_k(s,a)}} + 2\bar{D}\sqrt{5\log\left(\tfrac{16T^3}{\delta}\right)}\sum_k \sqrt{T_k} + K\bar{D}.$$

As for the analysis of UCRL2, we have that (cf. Eq. 20 of [9])

$$\sum_k \sum_{s,a} \tfrac{v_k(s,a)}{\sqrt{N_k(s,a)}} \ \leq\ \left(\sqrt{2}+1\right)\sqrt{S_\Sigma A T}.$$

Using that $\sum_k T_k = T$ together with Jensen's inequality, we obtain $\sqrt{T_k} \leq \sqrt{KT}$. Summarizing we obtain using the bounds on the number of episodes of Lemma 5 (c) after some simplifications and noting that $|\Phi| \leq S_\Sigma$ a regret bound of

$$const_1 \cdot D\sqrt{S_{\max} S_\Sigma A T \log\left(\tfrac{T}{\delta}\right)} + const_2 \cdot D\sqrt{S_\Sigma A T (\log T)\left(\log \tfrac{T}{\delta}\right)} + const_3 \cdot

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

291 the regret bounds for UCRL2 or a variation of it can be improved (this might be possible w.r.t. the
292 parameters $S$ and $D$, cf. [9]) this automatically gives improved bounds for a corresponding variant of
293 UCB-MS.

294 In [5], it has been tried to use some form of regularization so that models with large state space are
295 less appealing. However, this did not avoid the dependence of the claimed bounds on $S_\Sigma$. It is an
296 interesting question whether some improved regularization approach can give bounds that do only
297 depend on $S_{\phi^\circ}$. In general, the right dependence of regret bounds on the size of the model set $\Phi$ is
298 also an open problem.

299 Another question that is still open also for the MDP setting is whether the diameter can be replaced
300 by the *bias span* $\lambda^*$ of the optimal policy. With an upper bound on $\lambda^*$, one could replace UCRL2 by
301 the SCAL algorithm of [13]. However, the guessing scheme we employ for the diameter does not
302 work for SCAL, as chosen policies may not be optimistic anymore, if the guess for $\lambda^*$ is too small.

303 Another direction for future research are generalizations to infinite model sets, which for the case of
304 discrete sets has already been done for the BLB algorithm [6]. Parametric sets of models would be an
305 interesting next step from there.

306 A different question are approximate Markov models as considered in [14], where the assumption
307 that there is a Markov model is dropped. The results given there are also affected by the mentioned
308 error in the proof of the OMS regret bound. We think that our approach can be adapted, however the
309 details still have to be worked out.

## Footnotes

[1]The confidence intervals shown here are the ones we use in the following and slightly differ from the confidence intervals given for UCRL2 in [9]. That is, the confidence $\delta$ of the original values is replaced by $\delta/2t^2$ to guarantee smaller error probability, which is needed in our analysis.

[2]As noted before, the error probability $\delta$ has been changed from $\delta$ to $\delta/2t^2$ here.

[3]The bound in Lemma 3 is not explicitly stated for single episodes in [9] but easily follows from equations (8), (9), (15)–(17), and the argument given before equation (18), choosing confidence $\delta/t^2$ instead of $\delta$.

[4]As for UCRL2, we set the accuracy in episode $k$ to be $1/\sqrt{t_k}$.

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

# A  Proofs

## A.1  Proof of Theorem 10

The proof is like that for Theorem 6 only that now $S$ instead of $D$ is guessed and the comparison term $\Gamma_t$ is different. That is, any Markov model $\phi^\circ$ will not be discarded with high probability once $S \geq S_{\phi^\circ}$. Therefore, there will be at most $\lceil |\Phi| \log_2 S_{\phi^\circ} \rceil$ episodes that are terminated by discarding a model.

Let $S_k$ be the guess for the size of the state space in episode $k$. Then as in the proofs of Theorems 4 and 6, the regret in each episode $k$ can be shown to be bounded by $\Gamma_{t_k}(S_k) + 1$. As $S_k \leq 2S_{\phi^\circ}$, summing over all $\leq \lceil |\Phi| \log_2 S_{\phi^\circ} \rceil + S_\Phi A \log T$ episodes, Jensen's inequality gives the claimed regret bound. $\qquad\qquad\Box$

## A.2  Proof of Theorem 11

The proof is like that for Theorem 10. There will be at most $\lceil |\Phi| \log_2 (DS_{\phi^\circ}) \rceil$ episodes that are terminated by eliminating a model, while the regret in each episode $k$ is bounded by $\Gamma_{t_k}(\widetilde{DS}_k) + 1$, where $\widetilde{DS}_k \leq 2DS_{\phi^\circ}$ is the guess for episode $k$. A sum over the episodes gives the claimed bound. $\qquad\qquad\Box$