[Reviews · NeurIPS 2019]

Reviewer 1



The authors present a regret analysis for learning state representation. They propose an algorithm called UCB-MS with O(\sqrt{T}) regret, which improves over the currently best result. The paper is well-organized and easy to follow. The authors also explain the possible methods and directions to further improve the bound. The paper could be more clear if lemma 3 was proved in appendix. It may be obvious for someone who has read the similar proof. However, since it is crucial for the proof of the main theorem and is not explicitly stated in [9], explaining the steps can be helpful for a slightly more general audience. I wonder if there is really novel technique or idea in the paper. UCRL2, which is based on optimism in the face of uncertainty principle, chooses the optimistic model in plausible MDPs set in each episode. It is natural to choose optimistic representation model based on the same principle, which results in a regret bound relative to S_{\sum} instead of S in UCRL2 setting. Since there is not empirical result in the paper, I am not sure the significance of the result. For example, when solving RL problems such as Atari games, we may test different representation methods. If one of them approximately satisfy Markov property and achieve good performance, then we use this representation and the problem is solved. Are there some cases where we need to use algorithms to learn the best representation methods in a finite representation set? Besides, the regret can scale linearly in \sqrt{|\Phi|} since S_{\sum] scales linearly in \sqrt{|\Phi|} in worst case, which means the algorithm cannot perform well for large or even infinite representation set. Another tiny comment is that there are many algorithms such as UCBVI which enjoys square root dependence on S. Is it possible to use the framework and techniques of UCBVI to improve the dependence of the number of states? I believe the improvement of the dependence on S is important since S is always assumed to be extremely large. Overall, I believe the paper is correct and interesting, but I am not sure about the above issues. Maybe I will increase the score after rebuttal. ---------after rebuttal--------- I thank the authors for their response. The authors addressed my concerns in the feedback. Though I still believe that the algorithm is natural and simple since the framework is similar to UCRL2, the regret bound does improve over the currently known best result. Considering these factors, I updated my score to weak accept.

Reviewer 2



This work can be considered as an extension of UCRL to state representation tasks, where the agent needs to choose an optimal MDP mapping for the ordinary task first and then finds an optimal policy. The originality of this work is not significant, because it mainly extends UCRL to a similar setting. Still, I think it gives a new way and tighter bound for the state representation problem. The writing is clear and easy to understand. Some details I aim to concern: (1) Can UCB-MS ensure the final model to be a Markov model? If not, does the result still make sense? (2) Is there any discussion about the lower bound for this task? Since the regret bound of UCRL is near-optimal, I think the regret bound of UCB-MS should match lower bounds with respect to $T$. How about other parameters like $S$, $A$ or $\Phi$?

Reviewer 3



I see the main significance of this contribution in the following: a) it re-establishes a regret bound of O(sqrt (Phi * T)) which had previously been invalidated due to results in prior work having been shown as incorrect. It provides a somewhat simpler (subjectively) solution to the problem in terms of UCRL2. Regarding originality, this paper draws upon explored techniques, making this more of a refinement paper than a groundbreaking one, however, the novelty is still sufficient for a decent conference submission. The paper is very clearly written and easy to follow, and the ideas are presented in a natural way. I did not very the results thoroughly, but there to the extent that I did, there were no obvious red flags.

Reviewer 4



In this work the authors studied the online learning problem in finite -horizon RL setting when it also involves the learning of representation space. The paper is generally clearly-written, with several comparisons of performance with UCRL2, BLB and OMS discussed. Regarding originality and significance, I found the results of UCB-M novel. However, it seems to me that the results are extension of UCRL2 with an addition of representation learning. Also terms such as S_\Sigma in the regret bound in Theorem 4, which indicates the total size of state space of all models and comes from the representation learning part of the problem, may make the bound very loose. Therefore, I'm not really sure how useful the regret analysis is, without seeing some form of lower bound analysis (if possible) or empirical validation. Regarding the sections of unknown diameter, I appreciate the analysis on the doubling scheme for guessing D, and how its effect is compared with that in BLB. I also see that the authors are trying to address the aforementioned issue of S_\Sigma by introducing the notion of effective size S_\phi, but I am not sure how easy such a term can be derived for general representation classes. Can the authors provide some simple examples of S_Sigma and S_phi (or\min_\phi S_\phi) to illustrate their differences? Finally, the sections regarding the guessing of DS is good, but it seems to be a direct extension of the above results. In conclusion, I found the significance moderate, but given the low number of work in this field, I think the paper probably still provides some theoretical value to the online learning (for RL) community especially when the underlying Markov model assumption is relaxed and when representation learning is also taken into account.

[Author Response · NeurIPS 2019]

We would like to thank all reviewers for their insightful comments.

**Reviewer 1**

• **"I wonder if there is really novel technique or idea in the paper. (...) It is natural to choose optimistic**
**representation model (...)"**
We agree that our solution is simple and natural. The fact that previous approaches have used more complicated
algorithms to obtain worse bounds should be an indication that algorithm design and analysis are not obvious, however.

• **"(...) I am not sure the significance of the result. For example, when solving RL problems such as Atari**
**games, we may test different representation methods. If one of them approximately satisfy Markov property**
**and achieve good performance, then we use this representation and the problem is solved."**
In the mentioned scenario it has to be specified how to efficiently test (a) the (approximate) Markov property and (b)
good performance for each representation. (Note that for (b) you still have to find a good/optimal policy under each
representation.) If you have a simulator and you don't care about efficiency and cost of exploration (i.e., reward), you
can test (i.e., try to learn the optimal policy) using multiple models and then cross-validate the best. But if you care
about efficiency and overall performance, this is just not possible. Our approach shows that (a) is not necessary, while
for (b) we offer an explicit solution for the arising exploration-exploitation problem (for simultaneously choosing a
model and a policy) that is also efficient as the regret bound shows.

• **"Are there cases where we need to (...) learn the best representation methods in a finite representation set?"**
In many applications several sensory measures are available but it is not clear what is the most suited representation of
the system. Experts often have ideas about "reasonable" models and/or features combinations. One can also think of
using representations that worked well in other similar problem settings. The proposed algorithm may leverage this
information and quickly discard models that are not suited for the specific problem. Clearly, this is not the most generic
case but it is a first step toward having an efficient algorithm for selecting the representation.

• **"Besides, the regret can scale linearly in $\sqrt{|\Phi|}$ since $S_\Sigma$ scales linearly in $\sqrt{|\Phi|}$ in worst case, which means the**
**algorithm cannot perform well for large or even infinite representation set."**
While performance bounds will generally depend on the model space $\Phi$, it is an open problem whether the $\sqrt{|\Phi|}$-
dependence can be improved in the considered setting, cf. also our response to Reviewer 2. Still, there are techniques
that still work for infinite representation sets (see e.g. reference [6]) that will work for our approach as well.

• **"(...) there are many algorithms such as UCBVI which enjoys square root dependence on $S$. Is it possible to**
**use the framework and techniques of UCBVI to improve the dependence of the number of states?**
Please note that the regret bound for UCBVI has been derived for the simpler *episodic* setting. For the average reward
setting, it is still an open question whether $\sqrt{S}$-bounds are achievable. Our approach can be adapted to the episodic
case when the regret bounds would benefit from the improved bounds available in this setting. As discussed in the
paper, any optimistic algorithm with improved bounds could be used within our framework to obtain better bounds.

• **"The paper could be more clear if lemma 3 was proved in appendix."**
An explicit derivation of Lemma 3 would have taken a few pages mostly repeating material from [9], so we decided
not to include it in the paper. However, we will think about whether it is possible to give more details about the proof
without the need to copy content from [9].

**Reviewer 2**

• **"Can UCB-MS ensure the final model to be a Markov model? If not, does the result still make sense?"**
An important point of the paper is that for obtaining regret bounds in the online setting it is actually not necessary (and
in some cases not even possible) to identify the true (Markov) model. As long as a non-Markov model gives at least the
same reward that would be expected from a Markov model there is no need to discard it. Such a model could be e.g. a
good (non-Markovian) approximation. The regret is always measured with respect to the true Markov model however.

• **"Is there any discussion about the lower bound for this task? (...) I think the regret bound of UCB-MS should**
**match lower bounds with respect to $T$. How about other parameters like $S$, $A$ or $\Phi$?"**
The $\sqrt{A}$-dependence is optimal as for UCRL2, while the optimal dependence on $S$ is still an open question (also
for the MDP case). The optimal dependence on $|\Phi|$ in our setting is also open. The closest result we know is for
aggregation techniques with *full* information where it is possible to obtain $\log(|\Phi|)$. Obviously, in our setting we have
less information and it is not clear if it is possible to obtain logarithmic dependence.

**Reviewer 3**
Thank you very much for the positive feedback!

[Meta-Review · NeurIPS 2019]

This paper proposes a natural extension of UCRL2 to learning state representations. The proposed algorithm chooses optimistically over a finite set of candidate MDPs and their corresponding policies. The algorithm is analyzed and improves over existing regret bounds. The paper was discussed and all reviewers agree that this is a natural extension of UCRL2 that deserves to be published.